# Genetic Polymorphisms of *ALDH2* and *ADH1B* in Alcohol-Induced Liver Injury: Molecular Mechanisms of Inflammation and Disease Progression in East Asian Populations

**DOI:** 10.3390/ijms26178328

**Published:** 2025-08-28

**Authors:** Tomoko Tadokoro, Kyoko Oura, Mai Nakahara, Koji Fujita, Joji Tani, Asahiro Morishita, Hideki Kobara

**Affiliations:** Department of Gastroenterology and Neurology, Faculty of Medicine, Kagawa University, Kita-gun 761-0793, Kagawa, Japan; oura.kyoko@kagawa-u.ac.jp (K.O.); nakahara.mai@kagawa-u.ac.jp (M.N.); fujita.koji@kagawa-u.ac.jp (K.F.); tani.joji.kb@kagawa-u.ac.jp (J.T.); morishita.asahiro@kagawa-u.ac.jp (A.M.); kobara.hideki@kagawa-u.ac.jp (H.K.)

**Keywords:** *ALDH2*, *ADH1B*, alcohol, liver disease, polymorphism, East Asia, acetaldehyde, risk stratification

## Abstract

Alcohol-associated liver disease (ALD) is a major cause of liver-related mortality worldwide; however, only a subset of heavy drinkers develop progressive disease, suggesting a role for host genetics. In East Asian populations, functional polymorphisms in alcohol-metabolizing enzymes, such as alcohol dehydrogenase 1B (ADH1B) and aldehyde dehydrogenase 2 (ALDH2), are common and significantly affect acetaldehyde metabolism. ADH1B accelerates ethanol oxidation, whereas ALDH2 impairs acetaldehyde detoxification and increases oxidative stress, inflammation, and liver injury. Based on genotype combinations, individuals were stratified into five alcohol sensitivity groups with differing risks of cirrhosis and cancer. Although ALDH2 deficiency often suppresses alcohol intake via aversive reactions, paradoxically, continued drinking increases the risk of liver and gastrointestinal cancers. Genetic risk stratification may inform personalized prevention and precision of public health approaches. However, expansion of direct-to-consumer genetic testing has raised ethical and educational challenges. Understanding the interaction between alcohol metabolism and genetic variations is crucial for identifying high-risk individuals and guiding tailored interventions in East Asian populations.

## 1. Introduction

Alcohol-associated liver disease (ALD) is a leading cause of liver-related mortality worldwide and can lead to cirrhosis and hepatocellular carcinoma [1,2,3,4,5]. Only 35% of heavy drinkers develop advanced ALD, suggesting the involvement of additional contributing factors [6,7]. Several risk factors for ALD have been identified, including sex, obesity, drinking patterns, dietary factors, genetic predispositions unrelated to sex, and smoking [8]. In East Asia, genetic polymorphisms in the alcohol-metabolizing enzymes alcohol dehydrogenase 1B (*ADH1B*) and aldehyde dehydrogenase 2 (*ALDH2*) are highly prevalent, leading to substantial individual differences in drinking behavior and susceptibility to ALD [9]. High- and low-activity variants of alcohol dehydrogenase (*ADH*) and the low-activity variant of aldehyde dehydrogenase (*ALDH*) promote acetaldehyde accumulation, leading to a phenotype characterized by facial flushing and discomfort after alcohol consumption (Table 1) [10,11]. Acetaldehyde is crucial in the progression of ALD through mechanisms such as DNA damage, enhanced oxidative stress, and the induction of pro-inflammatory cytokines [12]. Furthermore, alcohol consumption disrupts the intestinal barrier, allowing bacterial endotoxins to translocate into the liver. This triggers hepatic immune dysregulation via Toll-like receptor 4 (TLR4) signaling, which is considered a central pathogenic mechanism driving chronic inflammation and fibrosis [13]. Individuals with *ALDH2* polymorphisms have a markedly reduced capacity to metabolize acetaldehyde, potentially leading to more complex and severe disease manifestations, such as malignant tumors, myocardial infarction, and dementia [14,15,16,17,18]. This review focuses on the genetic polymorphisms of *ADH1B* and *ALDH2* and outlines the molecular mechanisms underlying alcohol metabolism and hepatic inflammation. Furthermore, we discuss the unique genetic background and disease susceptibility observed in East Asian populations, emphasizing the potential applications in personalized medicine and public health interventions.

## 2. Methods

We retrieved published articles from PubMed and MDPI from peer-reviewed journals. The search was conducted using keywords related to ALD and genetic polymorphisms, such as alcohol, alcoholic hepatitis, cirrhosis, East Asian, *ALDH2*, *ADH1B*, acetaldehyde, and direct-to-consumer genetic testing (n = 3605). Following the initial search, the articles’ reference lists were reviewed, and potentially eligible articles were selected. We selected literature that could be viewed in full text. Articles that were not reported in English or in which the participant was diagnosed with a liver disease other than ALD were excluded. Quality assessment and data extraction were performed independently by two reviewers. In total, 82 papers were extracted.

## 3. Ethanol Metabolism in the Liver

Ethanol metabolism occurs primarily in the liver and involves a two-step oxidation process that converts ethanol to harmless acetate. Ethanol is oxidized to acetaldehyde by ADH, using NAD^+^ as a coenzyme and producing NADH. Although there are multiple isoforms of ADH, ADH1B is the principal enzyme responsible for ethanol metabolism in humans, and its enzymatic activity varies significantly depending on genetic polymorphisms [9,19]. The generated acetaldehyde is converted into acetate using ALDH2. ALDH2 is located in mitochondria and is important in the rapid detoxification of acetaldehyde. If ALDH2 activity is reduced, acetaldehyde accumulates in the body and exerts toxic effects [9]. Even small amounts of alcohol can trigger a strong “flushing response,” characterized by facial flushing, palpitations, and nausea.

ALDH2 and ADH1B are key enzymes involved in alcohol metabolism, and individual differences in their activities directly affect the pharmacokinetics of ethanol and acetaldehyde.

The genes encoding these enzymes have functional polymorphisms that are particularly prevalent in East Asian populations [3,7].

The progression of ALD is primarily driven by chronic inflammatory responses caused by the direct effects of ethanol and cellular damage induced by its metabolic byproducts [8,9].

### 3.1. The Direct Hepatotoxic Effects of Alcohol

#### 3.1.1. Increased Oxidative Stress

Under moderate alcohol consumption conditions, ethanol is primarily metabolized by ADH; however, excessive alcohol intake induces the expression of cytochrome P450 2E1 (CYP2E1), a microsomal drug-metabolizing enzyme [20]. During heavy alcohol consumption, if blood ethanol concentrations are elevated, ethanol is metabolized by CYP2E1, which has a higher Km than ADH, leading to the generation of free radicals and induction of cellular damage [21]. CYP2E1 metabolizes ethanol to acetaldehyde in an NADPH-dependent manner; however, it enhances NADPH oxidase activity, leading to increased production of reactive oxygen species (ROS), mitochondrial dysfunction, and ultimately cell death [21,22].

#### 3.1.2. Inhibition of Lipid Metabolism

The mechanisms by which excessive alcohol consumption leads to hepatic lipid accumulation include the suppression of gluconeogenesis due to increased redox potential (elevated NADH/NAD^+^ ratio) resulting from alcohol metabolism [20], reduced fatty acid β-oxidation through downregulation of PPARα expression [23], enhanced fatty acid and triglyceride synthesis via inhibition of AMP-activated protein kinase (AMPK) and the induction of SREBP1c [24], mobilization of free fatty acids from peripheral tissues [25], and impaired very-low-density lipoprotein (VLDL) secretion [24]. These processes, together with oxidative stress, lead to excessive lipid accumulation in the hepatic parenchyma, resulting in alcohol-induced fatty liver disease, which serves as a precursor to liver fibrosis, cirrhosis, and hepatocellular carcinoma (HCC).

#### 3.1.3. Disruption of the Intestinal Barrier and Translocation of Lipopolysaccharide to the Liver

Dysbiosis and impairment of the intestinal barrier function contribute to chronic liver disease through dysregulated gut–liver axis signaling [26,27,28,29]. Alcohol impairs intestinal barrier function, allowing lipopolysaccharides (LPS) derived from the gut microbiota to translocate into the bloodstream [30,31,32]. This LPS is recognized by Toll-like receptor 4 expressed on hepatic Kupffer cells, leading to their activation and the production of pro-inflammatory cytokines, such as TNF-α and IL-1β. These cytokines directly induce hepatocellular injury by triggering apoptosis and necrosis [27]. Activated Kupffer cells release pro-inflammatory cytokines and reactive oxygen species, such as superoxide and nitric oxide, leading to mitochondrial dysfunction and hepatocellular necrosis [33]. TNF-α, IL-1, and reactive oxygen species (ROS) released from Kupffer cells stimulate hepatic stellate cells, leading to increased expression of type I collagen and the promotion of fibrogenesis [34].

### 3.2. Hepatotoxic Effects of Acetaldehyde

#### 3.2.1. Direct Cytotoxicity to Hepatocytes

Acetaldehyde is highly reactive and binds to intracellular proteins, DNA, and lipids in hepatocytes, thereby impairing cellular function and leading to the necrosis and apoptosis of HCC. Acetaldehyde forms acetaldehyde–protein adducts that can trigger immune responses and promote inflammation [35,36].

#### 3.2.2. Increased Risk of Carcinogenesis

Acetaldehyde is classified as a carcinogen and increases the risk of various cancers, including liver cancer [37,38,39]. This is because acetaldehyde binds to DNA and proteins, leading to genetic mutations and abnormal cell proliferation [38,39,40,41]. Acetaldehyde can directly damage the DNA by the formation of mutagenic DNA adducts and interstrand crosslinks [42]. This disrupts normal DNA replication and induces genetic mutations. By binding to proteins, acetaldehyde interferes with normal cellular functions and signaling pathways, thereby contributing to the promotion of carcinogenesis [43].The International Agency for Research on Cancer classifies acetaldehyde associated with alcohol consumption as a group 1 carcinogen [44].

#### 3.2.3. Promotion of Fibrosis

Acetaldehyde induces the production of TGF-β1 and, through activation of the MAPK pathway and generation of reactive oxygen species (ROS), promotes the activation of hepatic stellate cells and gene expression of type I collagen [45,46,47]. The persistence of these processes contributes to the progression of liver fibrosis, ultimately leading to cirrhosis [47].

Inflammation in alcohol-associated liver disease is driven by multiple factors, including the disruption of the gut microbiota, accumulation of metabolic byproducts, activation of immune cells, and sustained release of cytokines. Acetaldehyde plays a central role in the intensity and persistence of this inflammatory response, and individual genetic backgrounds—particularly polymorphisms in *ALDH2* and *ADH1B*, which are involved in acetaldehyde metabolism—have a significant influence [48,49].

## 4. Impact of *ALDH2* and *ADH1B* Polymorphisms on Liver Disease Progression

### 4.1. Association Between ADH1B and Liver Disease

ADH1B converts ethanol to acetaldehyde. The *ADH1B*2* variant (Arg47His mutation) exhibits approximately 80–100-fold higher enzymatic activity than the low-activity **1* allele, thereby accelerating the conversion of ethanol to acetaldehyde [48]. The high-activity *ADH1B*2* variant is common among East Asians, including the Japanese and Chinese populations, with >80% of individuals carrying this allele [9]. Individuals with the *ADH1B*2* variant experienced a more rapid and pronounced increase in blood acetaldehyde levels. If they carry the low-activity form of ALDH2, acetaldehyde accumulation becomes more pronounced [48]. Because many unpleasant reactions after alcohol consumption are associated with the accumulation of acetaldehyde in the blood, individuals carrying the *ADH1B*2* allele may be more likely to avoid excessive alcohol intake, which may offer protection against alcohol use disorders [50]. Conversely, among individuals with alcohol dependence, the *ADH1B*2* allele is associated with an increased risk of liver cirrhosis, with an age-adjusted odds ratio for cirrhosis of 1.58 (95% confidence interval; 1.19–2.09) [51]. In the group with high-titer anti-HCV antibodies and the *ADH1B*2/*2* genotype, the adjusted odds ratio (AOR) for liver cirrhosis was 8.83 (95% confidence interval; 3.76–20.8) [51].

Although this may seem contradictory, individuals with alcohol dependence often carry the high-activity form of *ALDH2*, which enables the rapid breakdown of acetaldehyde and reduces unpleasant symptoms. In such cases, the *ADH1B*2* allele is believed to increase the risk of liver cirrhosis.

### 4.2. Association Between ALDH2 and Liver Disease

The *ALDH2**2 polymorphism (Glu504Lys mutation) is the most common variant in the *ALDH* gene family. It is virtually absent in individuals of European descent but is found in approximately 8% of the global population. Notably, approximately 40–50% of East Asians carry the *ALDH2*2* polymorphism, which results in markedly reduced enzymatic activity [52,53]. The low-activity variant of the *ALDH2* gene (*ALDH2*2*) results in an almost complete loss of enzymatic activity in homozygous individuals, whereas heterozygous carriers retain only approximately 10–20% of normal activity. Consequently, even small amounts of alcohol can trigger a strong “flushing reaction,” including facial flushing, palpitations, and nausea. In individuals with low ALDH2 activity, acetaldehyde detoxification is insufficient, leading to a stronger activation of the inflammatory pathways described earlier. However, because of the associated unpleasant symptoms, alcohol consumption tends to be self-limited; therefore, the low-activity ALDH2 variant is associated with a reduced risk of liver cirrhosis (OR = 0.78, 95% CI: 0.61–0.99) [49]. Therefore, *ALDH2* polymorphisms are considered an important protective factor against ALD in East Asian populations [49].

However, individuals with the active form of *ALDH2* (**1/*1* genotype) have been reported to have a higher risk of developing alcohol-related liver cirrhosis than those with the low-activity *1/*2 genotype (AOR = 1.43, 95% CI: 1.01–2.02) [49,51]. This is due to their ability to tolerate larger amounts of alcohol, leading to a greater chronic burden on the liver and a higher frequency of cirrhosis [51,54]. Recent studies have suggested that p53 suppresses ethanol-induced fatty liver by inhibiting the activity of ALDH2 [55]. On the other hand, some studies have shown that ALDH2 deficiency or genetic variants are risk factors for alcohol-induced gut dysbiosis and liver injury [56]. Gene therapy using adeno-associated virus to supplement the dysfunctional mitochondrial ALDH2 enzyme with a functional *ALDH2*1* allele has attracted attention as a means to restore ALDH2 enzymatic activity [57].The role of ALDH2 in ALD varies across studies, highlighting the need for further investigation.

Thus, genetic polymorphisms in *ADH1B* and *ALDH2* significantly influence individual alcohol metabolic capacity and susceptibility to alcohol-induced inflammation and liver injury (Figure 1). Generally, individuals with the *ADH1B**2 variant or the low-activity ALDH2 variant tend to consume less alcohol because of the toxic and unpleasant effects of acetaldehyde, which offers protection against alcohol dependence and alcohol-related hepatocellular carcinoma. However, if such individuals consume alcohol, they may be at increased risk of developing certain types of cancer [39,40]. It is difficult to predict disease development based on a single polymorphism; rather, a combination of genetic polymorphisms in *ADH1B* and *ALDH2* may be associated with various diseases [58].

### 4.3. Classification into Five Groups Based on Alcohol Sensitivity

The rate of acetaldehyde accumulation and clearance can be determined by combining the polymorphisms of the two enzymes ALDH2 and ADH1B. The degree of genetic regulation of alcohol sensitivity can be classified into five groups based on decreasing alcohol tolerance (Figure 2, Table 2) [41,43]:Group I: ALDH2*1/*1 and ADH1B*1/*1;Group II: ALDH2*1/*1 and ADH1B*1/*2 or *2/*2;Group III: ALDH2*1/*2 and ADH1B*1/*1;Group IV: ALDH2*1/*2 and ADH1B*1/*2 or *2/*2;Group V: ALDH2*2/*2 and ADH1B*1/*1, *1/*2, or *2/*2.

The *ALDH2*2*allele plays a role in suppressing alcohol consumption, whereas the *ADH1B*1*allele is associated with promoting alcohol consumption [59].

#### 4.3.1. Group I: *ALDH2*1/*1* (Active Type) + *ADH1B*1/*1* (Low-Activity Type)

The group with the lowest alcohol sensitivity, i.e., the most tolerant of alcohol, had the highest risk of depression, anxiety, and alcohol dependence [58,60]. With this combination, acetaldehyde is produced slowly and rapidly, making it less likely to accumulate in the body and triggering a flushing reaction. Consequently, individuals tend to consume larger amounts of alcohol and are more likely to maintain heavy drinking over time. Consequently, chronic alcohol exposure persists in the liver and increases the risk of liver cirrhosis [51].

#### 4.3.2. Group II: *ALDH2*1/*1* (Active Type) + *ADH1B*1/*2* or **2/*2* (High-Activity Type)

The low-to-moderate-sensitivity group is characterized by efficient acetaldehyde metabolism and a lower likelihood of experiencing unpleasant symptoms during alcohol consumption. Consequently, individuals tend to consume larger amounts of alcohol and may unknowingly continue high-risk drinking, which may increase their risk of liver cirrhosis [61]. This group is believed to have an increased risk of alcohol-related liver cirrhosis owing to its synergistic effect with hepatitis C infection [51]. It was also associated with an increased risk of colorectal cancer (OR = 1.35; 95% CI: 1.11–1.63) [62].

#### 4.3.3. Group III: *ALDH2*1/*2* (Heterozygous) + *ADH1B*1/*1* (Low-Activity Type)

In the moderate-to-high alcohol sensitivity group, acetaldehyde production is slow, but its breakdown is significantly delayed, leading to the sustained accumulation of acetaldehyde after alcohol consumption. This genotype combination occurs in approximately 3–5% of the East Asian population (Japan, China, and Korea) [9]. Among drinkers, the risk of esophageal cancer is increased (OR = 3.02, 95% CI: 1.54–5.91), as is the risk of cancers of the pharynx, larynx, and nasal cavity (OR = 1.56, 95% CI: 1.20–2.02) [63]. It is associated with an increased risk of bladder cancer (OR = 4.00, 95% CI: 1.81–8.87, *p* = 0.001) [64]. In Vietnam, this combination has been reported to be associated with an increased risk of alcohol-related liver cirrhosis [61].

#### 4.3.4. Group IV: *ALDH2*1/*2* (Heterozygous) + *ADH1B*1/*2* or **2/*2* (High-Activity Type)

In the high-alcohol sensitivity group, acetaldehyde is rapidly produced but slowly breaks down, thereby rapidly increasing blood acetaldehyde levels. Even small amounts of alcohol can easily trigger a “flushing reaction,” including facial flushing, palpitations, nausea, and headaches. This genotype combination is found in approximately 30–50% of the East Asian population [52,53]. It is a risk factor for malignancies of the oral cavity, pharynx, larynx, and esophagus (OR = 10.31, 95% CI: 5.45–18.85) [43,48,49]. No significant increase in risk was observed among never or infrequent drinkers [59]. The combination of *ALDH2*1/*2* (heterozygous) and *ADH1B*1/*2* is considered a risk factor for smoking [65].

In individuals with genotypes that lead to acetaldehyde accumulation, the risk of liver cirrhosis is lower [61], whereas some studies have suggested a higher risk of hepatocellular carcinoma [66]. However, these findings are inconsistent across studies.

#### 4.3.5. Group V: *ALDH2*2/*2* (Homozygous) + Any *ADH1B* Genotype (**1/*1*, **1/*2*, or **2/*2*)

This is the group with the highest alcohol sensitivity (strong flushing reaction). Regardless of the *ADH1B* genotype, acetaldehyde accumulates almost completely in the body, and even small amounts of alcohol trigger a strong flushing reaction. The prevalence varies by study; however, approximately 1–8% of the East Asian population falls into this category [9].

Alcohol consumption in this group is generally low, which helps prevent the development of alcohol dependence [67], and considered protective against alcohol-related liver disease [54,61,68].

Continued alcohol consumption in individuals with ALDH2-deficient hepatocytes leads to the excessive production of harmful oxidized mitochondrial DNA via extracellular vesicles, which has been associated with an increased risk of alcohol-related HCC through fibrosis in patients and mouse models [69,70]. Furthermore, in heavy drinkers, individuals carrying the *ALDH2* Lys allele had an odds ratio of 3.57 (95% CI: 2.04–6.27) for gastric cancer compared with Glu/Glu carriers [71].

The impact of *ALDH2*2* varies depending on the stage of alcohol involvement. Although it may be protective against the progression to heavy drinking, it can become a risk factor under conditions of sustained or excessive alcohol consumption. The association between the *ALDH2* and *ADH1B* genotypes and cancer risk varies across studies, and stratification based on alcohol consumption status is essential to clarify these discrepancies. Genetic polymorphisms in *ALDH2* and *ADH1B* are key determinants of individual alcohol sensitivity and are significant in the progression and natural history of liver disease. Risk stratification based on these genetic factors is becoming increasingly necessary for the prevention and early intervention of liver disease.

On the other hand, as previously described, the *ALDH2*2* allele (inactive type) is highly prevalent in East Asia (particularly China, Korea, and Japan), but it is rarely observed in other regions such as Europe, Africa, the Middle East, and the Americas [72]. Large-scale epidemiological studies conducted in East Asian populations have demonstrated that the *ALDH2*2* allele markedly increases the risk of esophageal and head and neck cancers; however, in European and African populations, where the allele frequency is extremely low, sufficient investigation has not been conducted. The *ADH1B**2 allele (fast-metabolizing type) is observed at a very high frequency in East Asia, whereas its frequency is very low in Europe (approximately 6%) and somewhat higher in African populations (approximately 19%). It has been reported to exert a protective effect against alcohol dependence [73]. Since the active *ADH* polymorphism observed in 70% of East Asians and 6% of Europeans is *ADH1B*2* allele, whereas the variant found in 19% of Africans is ADH1B3, a simple comparison between populations is difficult [73]. The grouping based on genetic polymorphisms shown in Table 2 has been frequently reported in East Asian populations; however, in other regions, particularly regarding *ALDH2* polymorphisms, sufficient validation has not been conducted, and further studies are needed to determine its applicability.

## 5. Risk Stratification and Preventive Guidance Based on Genetic Polymorphisms

Direct-to-consumer (DTC) genetic testing refers to genetic tests sold directly to consumers without the involvement of healthcare providers [74]. It is attracting attention as a tool for preventive public health education [75]. Despite the absence of symptoms, knowing one’s genetic profile can lead to increased risk awareness, such as recognizing that “alcohol consumption may be harmful for me,” which may promote behavioral changes, such as reduced alcohol intake and increased willingness to seek medical attention. In prospective studies, the *ALDH2*2* genetic polymorphism was shown to reduce the risk of alcohol-related problems among Asian university students [75,76]. Recognizing individual genetic polymorphisms at a young age using DTC testing may contribute to the prevention of various alcohol-related problems. Several concerns are associated with DTC testing [77,78]. DTC testing does not provide the genetic counseling necessary for consumers to interpret the results and make informed decisions based on them [74,79]. Consequently, there is a risk of misunderstanding or excessive anxiety [78,80]. The general public shows strong interest in receiving genetic feedback related to outcomes, such as alcohol dependence and mental health; however, there may be a lack of understanding of its implications [81]. Personalized healthcare integrated with smartphones and digital tools is in increasing demand [78,82]. Moving forward, it will be important for healthcare professionals with accurate knowledge to utilize digital tools and other means to provide educational activities based on the information accessible to the public.

## 6. Conclusions and Future Directions

Genetic polymorphisms in *ALDH2* and *ADH1B* significantly influence alcohol metabolism and the accumulation of toxic metabolites, thereby determining the risk of various alcohol-related diseases, including liver disease. These genetic variants are common among East Asian populations, making personalized prevention strategies and medical interventions particularly important. Early identification of individuals with high alcohol sensitivity based on their genotype allows for tailored advice and guidance regarding drinking behavior (Figure 1). In particular, individuals carrying the low-activity *ALDH2*2* variant are prone to acetaldehyde accumulation, even with small amounts of alcohol, placing them at an increased risk of esophageal cancer, liver cirrhosis, and head and neck cancers. Alcohol restriction and targeted cancer screening may be beneficial in high-risk individuals.

However, caution is warranted when interpreting the results of DTC genetic testing, as genetic factors alone cannot fully predict the risk of alcohol-related diseases. Misinterpretation of genetic predisposition may result in inappropriate drinking behaviors. Therefore, it is essential to establish systems that provide professional genetic counseling and guidance.

Thus, there is a growing need to integrate *ALDH2* and *ADH1B* polymorphism data into precision prevention strategies aligned with personalized medicine and broader population-level approaches known as precision public health. Through genotype-based risk stratification, the health risks associated with alcohol consumption can be minimized, contributing to primary prevention and early detection of the disease (Figure 2).

## Figures and Tables

**Figure 1 ijms-26-08328-f001:**
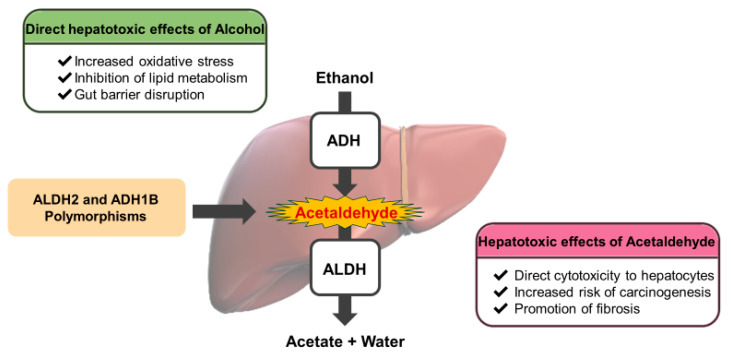
Alcohol metabolism and mechanisms of alcohol-induced liver injury associated with genetic polymorphisms in alcohol-metabolizing enzymes. Ethanol is metabolized in the liver via a two-step enzymatic process. Ethanol is oxidized to acetaldehyde by alcohol dehydrogenase (ADH), and acetaldehyde is detoxified to acetate and water by ALDH. Genetic polymorphisms in *ADH1B* and *ALDH2* significantly influence the activity of these enzymes, resulting in variable accumulation of acetaldehyde and individual differences in alcohol sensitivity and disease risk.

**Figure 2 ijms-26-08328-f002:**
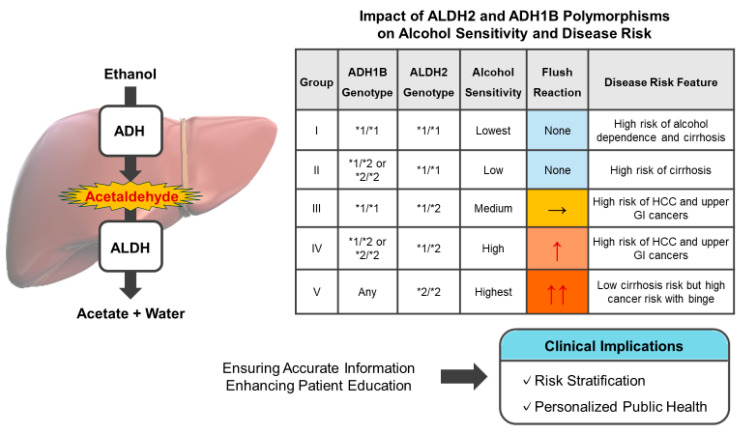
Genetic polymorphisms in alcohol metabolism and their clinical implications. The table classifies individuals into five groups based on their *ADH1B* and *ALDH2* genotypes, alcohol sensitivity, and flushing response. These groupings correlate with specific clinical risks, such as alcohol dependence, cirrhosis, and upper gastrointestinal (GI) cancers. Alcohol-induced hepatic damage is mediated by the direct hepatotoxic effects of ethanol, the accumulation of acetaldehyde, and disruption of the gut barrier. Understanding these genotype-based differences can inform risk stratification and promote personalized public health interventions, particularly in East Asian populations in which *ALDH2* polymorphisms are highly prevalent. → represents a moderate flushing response, ↑ represents a severe flushing response, and ↑↑ represents a very severe flushing response.

**Table 1 ijms-26-08328-t001:** Functional polymorphisms of *ADH1B* and *ALDH2* and their impact on enzymatic activity.

Gene	SNP (Amino Acid Change)	Genotype	Enzymatic Activity	Characteristics
*ADH1B*	rs1229984 (Arg47His)	*1/*1 (Arg/Arg)	Low	Slow conversion of ethanol to acetaldehyde
		*1/*2 (Arg/His)	Intermediate–High	Increased enzymatic activity
		*2/*2 (His/His)	High	Rapid production of acetaldehyde
*ALDH2*	rs671 (Glu504Lys)	*1/*1 (Glu/Glu)	Active (normal)	Normal conversion of acetaldehyde to acetate
		*1/*2 (Glu/Lys)	Low	Approximately 10–20% activity; associated with facial flushing, etc.
		*2/*2 (Lys/Lys)	Inactive (deficient)	Near-zero activity; strong alcohol intolerance

**Table 2 ijms-26-08328-t002:** Grouping based on alcohol sensitivity. HCC, hepatocellular carcinoma; GI, gastrointestinal.

Group	*ADH1B* Genotype	*ALDH2* Genotype	Frequency	Alcohol Sensitivity	Flush Reaction	Disease Risk Feature
I	*1/*1	*1/*1	5–10%	Lowest	None	High risk of alcohol dependence and cirrhosis
II	*1/*2 or *2/*2	*1/*1	50%	Low	None	High risk of cirrhosis
III	*1/*1	*1/*2	3–5%	Medium	Mild	High risk of HCC and upper GI cancers
IV	*1/*2 or *2/*2	*1/*2	30–50%	High	Strong	High risk of HCC and upper GI cancers
V	Any	*2/*2	1–8%	Highest	Very Strong	Low cirrhosis risk but high cancer risk with binge

## Data Availability

Not applicable.

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
