# Peer review of "Genetic Polymorphisms of *ALDH2* and *ADH1B* in Alcohol-Induced Liver Injury: Molecular Mechanisms of Inflammation and Disease Progression in East Asian Populations"

_ijms, 2025, doi:10.3390/ijms26178328_

Round 1

Reviewer 1 Report

Comments and Suggestions for Authors

General comments

The review manuscript in question deals with the role of host genetics in the Alcohol-associated liver disease (ALD) with a specific focus on the impact on East Asian populations where functional polymorphisms in alcohol-metabolizing enzymes, such as alcohol dehydrogenase 1B (ADH1B) and aldehyde dehydrogenase 2 (ALDH2), are quite common by significantly affecting acetaldehyde metabolism. Genotype combinations are employed to stratify individuals into five alcohol sensitivity groups with differing risks of cirrhosis and cancer. Legitimately, the authors claim identifying high-risk individuals may help guiding tailored interventions in East Asian populations. The review is timely, well-written, and I think it will prove useful to the cause. There are only a few small things I would like to add, in my opinion.

Specific comments

1) There is no mention of alcohol-induced fatty liver disease, a condition that, through impaired ROS management, leads to overaccumulation of lipids in the liver parenchyma, leading to fibrosis, steatohepatitis/cirrhosis, and hepatocellular carcinoma. This component must be included and treated in accordance with the proposed stratification.

2) A few more figures with more pathogenetic details wouldn't hurt.

Author Response

Thank you very much for taking the time to review our manuscript. We fully agree with your comments and have revised the manuscript accordingly. We are submitting the revised version, in which all corrections related to Reviewer 1 are highlighted in blue.

1) There is no mention of alcohol-induced fatty liver disease, a condition that, through impaired ROS management, leads to overaccumulation of lipids in the liver parenchyma, leading to fibrosis, steatohepatitis/cirrhosis, and hepatocellular carcinoma. This component must be included and treated in accordance with the proposed stratification.

→Thank you for your comment. In accordance with your suggestion, we have added a reference to alcohol-induced fatty liver disease in section 3.1.2.

2) A few more figures with more pathogenetic details wouldn't hurt.

→Thank you for your suggestion. We have added Figure 1 illustrating the mechanisms of alcoholic liver injury and the role of genetic polymorphisms.

Reviewer 2 Report

Comments and Suggestions for Authors

The authors review the polymorphisms associated with ALD. They promise to discuss the molecular mechanisms of inflammation and its progression. While the work is well written and has an appropriate narrative, it falls short of its promise and does not provide an extensive analysis of either the molecular mechanisms or the progression. Other aspects that should also be improved include:

line 120 HCC cells?

In 3.2.2 more information about the relationship among acetaldehyde and carcinogenesis should be discussed.

Line 133-138, references are missed

The authors repeatedly mention HCC, but only in line 262 do they introduce the meaning correctly. They should correct and review the text to ensure there are no more errors of this type.

line 263 the OR increases to 3.57 (significantly) but relative to what value?

Line 275: DTC? (correctly describe acronyms)

line 289-291, the idea needs to be finalized

Figure 1 is not reflected in the body of the manuscript. In addition, it has low definition.

Line 317 to 320 is confusing and does not follow a continuous thread.

Author Response

Thank you very much for taking the time to review our manuscript. We fully agree with your comments and have revised the manuscript accordingly. We are submitting the revised version, in which all corrections related to Reviewer 2 are highlighted in yellow.

line 120 HCC cells?

→Thank you for your comment. It was a mistake; I meant HCC. I have corrected it.

In 3.2.2 more information about the relationship among acetaldehyde and carcinogenesis should be discussed.

→Thank you for the important comment. I have added a description regarding acetaldehyde and carcinogenesis in section 3.2.2.

Line 133-138, references are missed

→Thank you for the valuable comments. We have added the appropriate references.

The authors repeatedly mention HCC, but only in line 262 do they introduce the meaning correctly. They should correct and review the text to ensure there are no more errors of this type.

→Thank you for your suggestion. We have reviewed and revised the use of abbreviations, including HCC, throughout the manuscript.

line 263 the OR increases to 3.57 (significantly) but relative to what value?

→Thank you for your important comment. The value represents the odds ratio indicating how many times higher the risk of disease is for individuals with the ALDH2 Lys allele who engage in heavy drinking, compared with those with ALDH2 Glu/Glu who also engage in heavy drinking. We have revised the description according to your suggestion.

Line 275: DTC? (correctly describe acronyms)

→Thank you for your comment. DTC refers to Direct-to-consumer (genetic testing). I have revised the description accordingly and also reviewed and corrected the use of abbreviations throughout the manuscript.

line 289-291, the idea needs to be finalized

→Thank you for your suggestion. We believe that, going forward, it will be important for healthcare professionals with accurate knowledge to utilize digital tools and other means to provide educational activities based on the information accessible to the general public. We have added this sentence.

Figure 1 is not reflected in the body of the manuscript. In addition, it has low definition.

→Thank you for pointing this out. We have now referred to Figure 1 in the main text and replaced it with a higher-resolution version to improve clarity.

Line 317 to 320 is confusing and does not follow a continuous thread.

→Thank you for your valuable comment. We have revised the sentence to improve clarity and logical flow. The revised text now reads as follows:

“However, caution is warranted when interpreting the results of DTC genetic testing, as genetic factors alone cannot fully predict the risk of alcohol-related diseases. Misinterpretation of genetic predisposition may result in inappropriate drinking behaviors. Therefore, it is essential to establish systems that provide professional genetic counseling and guidance. “

Reviewer 3 Report

Comments and Suggestions for Authors

The manuscript by Tomoko Tadokoro et al. is aimed at analyzing genetic polymorphisms of ALDH2 and ADH1B genes in alcohol-induced liver injury in East Asian Populations. Although the research has been quite significant, I believe that the manuscript needs thorough revision and the text in the current version is not ready for publication and is of only very limited interest.

  1. The authors state in the title that they focused on the analysis of East Asian Populations, however, in the Methods section, where the authors indicated the criteria and keywords they used for the search, there is not a word about East Asian Populations. Thus, the content of the article does not correspond to the title. And either the title or the Methods section needs to be changed.
  2. Fig. 1. The font inside the figure is so small that it is unreadable. Please revise the Fig. 1
  3. The review is poorly illustrated; in particular the only figure in the manuscript does not reflect either the mechanism of susceptibility to the liver complications among population groups or the contribution of particular polymorphisms. More illustrations on the mechanisms of the contribution of polymorphisms to liver complications should be added, especially since even the title contains "Molecular Mechanisms"
  4. The authors divided the population into 5 different groups based on their own criteria, but it is not clear from the text how subjective or objective this method of population stratification is. In particular, the article sorely lacks a comparison with other populations, at least very briefly, in particular, is the division into groups done in other populations in the same way?
  5. The following comment also concerns the issue of comparability of the manuscript results with relevant studies on other populations. For example, what is the frequency of the identified polymorphisms in other populations, is there a similar relationship between particular SNPs and complications, cirrhosis, cancer, etc. Thus, this manuscript is very much in need of an additional chapter to discuss the findings, etc.

Author Response

Thank you very much for taking the time to review our manuscript. We fully agree with your comments and have revised the manuscript accordingly. The corrections related to Reviewer 3 are highlighted in green in the revised version, which we are submitting.

The authors state in the title that they focused on the analysis of East Asian Populations, however, in the Methods section, where the authors indicated the criteria and keywords they used for the search, there is not a word about East Asian Populations. Thus, the content of the article does not correspond to the title. And either the title or the Methods section needs to be changed.

→Thank you for your comment. Our study did include East Asian populations; however, this was inadvertently omitted from the Methods section. We apologize for the confusion. In response to your suggestion, we have revised the Methods section to clearly state that East Asian populations were included in the analysis.

Fig. 1. The font inside the figure is so small that it is unreadable. Please revise the Fig. 1

The review is poorly illustrated; in particular the only figure in the manuscript does not reflect either the mechanism of susceptibility to the liver complications among population groups or the contribution of particular polymorphisms. More illustrations on the mechanisms of the contribution of polymorphisms to liver complications should be added, especially since even the title contains "Molecular Mechanisms"

→Thank you for the valuable suggestion. We revised the figures by separating the mechanisms of hepatic complications and the genetic polymorphisms to enhance clarity. In addition, we increased the font size to improve visibility.

The authors divided the population into 5 different groups based on their own criteria, but it is not clear from the text how subjective or objective this method of population stratification is. In particular, the article sorely lacks a comparison with other populations, at least very briefly, in particular, is the division into groups done in other populations in the same way?

The following comment also concerns the issue of comparability of the manuscript results with relevant studies on other populations. For example, what is the frequency of the identified polymorphisms in other populations, is there a similar relationship between particular SNPs and complications, cirrhosis, cancer, etc. Thus, this manuscript is very much in need of an additional chapter to discuss the findings, etc.

→Thank you for your valuable comment. This classification has been reported in several studies from East Asia and was not originally developed by us1,2). We have revised the manuscript to clarify the prevalence and implications of ALDH2 and ADH1B polymorphisms in different populations. In particular, since the ALDH2 polymorphism is largely restricted to East Asia, we consider that this classification should be applied specifically to East Asian populations.

1)Yoshimasu, K.; Mure, K.; Hashimoto, M.; Takemura, S.; Tsuno, K.; Hayashida, M.; Kinoshita, K.; Takeshita, T.; Miyashita, K. Genetic alcohol sensitivity regulated by ALDH2 and ADH1B polymorphisms is strongly associated with depression and anxiety in Japanese employees. Drug Alcohol Depend 2015, 147, 130-136, DOI:10.1016/j.drugalcdep.2014.11.034.

2) Owaki, Y.; Yoshimoto, H.; Saito, G.; Dobashi, S.; Kushio, S.; Nakamura, A.; Goto, T.; Togo, Y.; Mori, K.; Hokazono, H. Effectiveness of genetic feedback on alcohol metabolism to reduce alcohol consumption in young adults: an open-label randomized controlled trial. BMC Med 2024, 22, 205, DOI:10.1186/s12916-024-03422-y.

Round 2

Reviewer 3 Report

Comments and Suggestions for Authors

The authors have responded to all my questions satisfactorily.